# Competitive Match Running Speed Demands and Impact of Changing the Head Coach in Non-League Professional Football[note 1]

**DOI:** 10.3390/s25092865

**Published:** 2025-04-30

**Authors:** Daniel T. Jackson, Richard C. Blagrove, Peter K. Thain, Anthony Weldon, Cain C. T. Clark, Adam L. Kelly

**Affiliations:** 1Research for Athlete and Youth Sport Development (RAYSD) Lab, Faculty of Health, Education and Life Sciences, Birmingham City University, Birmingham B15 3TN, UK; peter.thain@bcu.ac.uk (P.K.T.); anthony.weldon@bcu.ac.uk (A.W.); cain.clark@bcu.ac.uk (C.C.T.C.); 2School of Sport, Exercise and Health Sciences, Loughborough University, Loughborough LE11 3TU, UK; r.c.blagrove@lboro.ac.uk; 3Aston Villa Foundation, Aston Villa Football Club, Birmingham B6 6HE, UK

**Keywords:** soccer, sub-elite, external load, coaching

## Abstract

**Highlights:**

**What are the main findings?**
Positional roles influence match running speed demands in professional Non-League Football, with wide defenders and midfielders covering greater distances.Head Coach changes significantly reduce running performance but do not improve match outcomes.

**What is the implication of the main finding?**
Understanding running speed demands in professional NLF is key to informing training practices, player conditioning, and tactical strategies tailored to this level.

**Abstract:**

Match running speed demands vary across competitive levels of football, influenced by player position, tactical considerations, and Head Coach changes. In England, the level directly below professional football, Non-League Football (NLF), comprises full-time and part-time clubs. However, the running speed demands of professional teams at this level remain unknown. Therefore, this study aimed to investigate (1) the match running speed demands in a professional NLF team, and (2) the impact of changing the Head Coach on these physical demands. Match running speed data were collected via Polar Team Pro global positioning system (GPS) devices during 41 matches of a tier 6 NLF team, comprising 311 observations of 22 full-time outfield players. Linear mixed-effect models examined the relationship between running speed metrics and fixed effects of a Head Coach change (n = 3), player position, and match outcome, with match number as a random effect. The team average total distance (TD) was 10,479 ± 42 m, and high-speed running and sprinting were 431 ± 62 m and 99 ± 26 m, respectively. The results showed significant positional differences, with wide defenders and midfielders associated with a greater TD than central defenders and strikers. Moreover, a change in Head Coach was significantly associated with a reduced TD, and a similar downward trend was observed across other running speed metrics. The TD and positional differences observed are comparable with other football cohorts, yet HSR and sprinting distances were notably lower in professional NLF. The findings highlight NLF clubs’ challenges in transitioning to higher competitive levels and provide insights for performance and training. Further research is warranted to explore the influence of running speed demands, technical and tactical factors, and other determinants on success in NLF.

## 1. Introduction

Football is a highly competitive sport that requires players to repeatedly perform high-speed running efforts throughout a 90 min match [1]. Professional football clubs are usually confined to the top leagues within a nation. In English men’s football, these consist of the Premier League (PL; tier 1) and English Football League (EFL; tiers 2–4) [2]. The leagues below these are usually semi-professional or amateur, forming the National League System (tiers 5+), called Non-League Football (NLF). Tier 5 clubs incur an average deficit of GBP 1.1 million per season [3]. Consequently, the sports science and medicine practices at this level differ considerably from professional clubs due to funding constraints [4]. Adding to this complexity, higher tiers in NLF (i.e., tiers 5–6) contain both full-time and part-time clubs, creating a unique cohort of non-league professional clubs approaching the professional level. However, the literature is limited on professional clubs operating at this level in England, with the existing studies focusing on those in tiers 1–4 and semi-professional NLF clubs [5,6].

The growing use of player tracking technology, such as global positioning systems (GPSs), facilitates precise measurements of player movement during football matches and is increasingly common in NLF [4,7]. The Polar Team Pro (Polar Electro, Kempele, Finland) system provides relatively inexpensive access to a GPS and accelerometer unit, with a sampling rate of 10 Hz, for capturing match physical performance data (i.e., total distance covered) [8]. These data may be used for performance analysis or to inform on injury risk within professional contexts [9]. However, the utility and application of GPS data and performance analysis within NLF remains unclear and may vary from professional contexts due to staffing and financial constraints [4]. Professional players complete high-intensity (>19.8 km·h^−1^) running distances (HIRDs) of approximately 760 m and 200 m of sprinting (>25.1 km·h^−1^) per game [10]. The running speed demands of a football match, notably the TD and HIRD, are different across playing levels, with higher-level professional players covering greater distances compared to lower-level counterparts [11,12,13,14]. These differences in running speed demands may be due to variations in players’ physical characteristics. For example, GPS running performance data are associated with physical components, such as lower limb power, body composition, and aerobic and anaerobic capacities, which also differentiate elite from sub-elite players [15,16,17]. As NLF clubs can be professional or semi-professional, there may be differences in training time and players’ off-field commitments (e.g., employment other than football), potentially influencing players’ aerobic and anaerobic capacities and, consequently, running speed demands compared to semi-professional teams.

During a men’s professional competitive match, players typically cover total distances (TDs) of 9–14 km [11,18,19]. However, the running demands of a football match vary substantially, influenced by playing level, playing position, training load, technical proficiency, match location, and/or match tactics [18,20,21,22,23,24]. A better understanding of these physical requirements during a football match is important, as it allows for the effective planning and delivery of training to meet such demands, while informing injury preventative strategies [25]. Although the running speed demands of professional men’s football are well documented, these demands at the level directly below (i.e., Non-League Football, from tiers 5–6 in England) remain unclear.

A team’s technical and tactical performance is associated with the running speed profiles observed within a football match [26]. For instance, in English professional football, PL clubs, which contain the most technically proficient players, performed less high-speed running and sprinting distances than in lower leagues (Championship and League 1) [27]. Similarly, within the same league, more successful teams completed lower HIRD than less successful teams [28]. However, translating physical performance into match success remains challenging, as Head Coaches must implement tactical strategies and manage players in an unpredictable environment [29]. Additionally, the lower the league, the less likely the Head Coaches’ decision-making will be informed by objective data or supported by a sports scientist due to the lack of funding, resources, and expertise available [4].

In lower leagues, where differences in team quality are less pronounced (e.g., tier 5), the ability to predict success is more challenging than in higher leagues [30]. When a team performs poorly, or their results jeopardise their league position, club directors may change the Head Coach to instil performance improvements [31]. Introducing a new coach has been described as having a ‘shock effect’, which aims to motivate and improve performance [32]. Such changes have been reported to influence match outcomes and success within a poorly performing club [33]. However, the unpredictability in lower leagues may contribute to the turnover rates of Head Coaches. For example, during the 2023/24 season alone, 28 coach changes occurred in the National League (tier 5) compared to 11 in the Premier League (tier 1) (data from transfermarkt.co.uk, accessed 24 January 2025).

A change in Head Coach can also affect the running speed demands during matches, with short term increases in running performance attributed to the motivational improvement of the team [32,33]. Players have been shown to perform more accelerations and decelerations throughout a game under a new coach [23]. However, these effects are often short-lived, and there is uncertainty about whether a change in Head Coach will influence overall long-term seasonal success [34]. Within NLF, a high turnover of support staff (e.g., coaching, sports science, and medicine) has been reported, driven by financial constraints and team dynamics [4]. While coaching changes in professional football have been shown to produce short-term improvements in performance and running demands [23], it is unclear whether a similar trend occurs in NLF.

Given the uncertainties surrounding the impact of coach change in NLF and the potential disparities in running speed demands between professional and semi-professional clubs, it is important to understand these aspects of NLF better. While running speed demands across professional leagues in English football vary considerably [13,27], there are currently no data for professional NLF. Therefore, this study aimed to investigate the competitive match running speed demands in a professional NLF team, and whether these demands were impacted by a change in Head Coach during the season.

## 2. Materials and Methods

### 2.1. Experimental Design

This was a longitudinal study conducted across an entire season, examining running speed demands of a professional NLF team competing in National League North (tier 6) in the 2018–2019 season. Match running speed data were collected for 41 competitive league matches, using GPS wearable devices, with only the full-match observations included in the analysis. The results of preseason testing, independent of this current study’s data collection, are included to support the interpretation of match running speed demands.

### 2.2. Study Population

Twenty-two professional (i.e., full-time) first-team outfield players (age 22.8 ± 4.7 years, height 183 ± 7.5 cm, body mass 77.8 ± 8.1 kg), from a professional tier 6 NLF team, participated in the study. Clubs competing in NLF are not restricted to transfer windows, resulting in player roster changes throughout the season. Inclusion criteria were as follows: (a) an outfield player contracted at the beginning of the season, (b) players who joined the club during the season under contract, and (c) ≥18 years of age. Exclusion criteria for the study were as follows: (a) <18 years of age and (b) a player on trial at the club. All participants consented to use the GPS data for this study, which was granted ethical approval by the Birmingham City University Ethics Committee (9377).

Participants in this study typically completed four training sessions (i.e., 90–180 min per session), excluding gym or rehabilitation sessions, on a one match-day week. Player positions were assigned by the Head Coach, utilising a 3-5-2 tactical formation predominantly throughout the season. These positions were defined as central strikers (STs) (n = 4), midfielders (MFs) (n = 7), wide defenders (WDs) (n = 4), and central defenders (CDs) (n = 7). There were a total of three Head Coaches during the study period who managed a total of 41 competitive matches during the season, with the first Head Coach (Coach A) managing 26 matches, the second (Coach B) as interim Head Coach managing 4 matches, and the third (Coach C) managing 11 matches.

In professional football, both ${\dot{V}O}_{2\max}$ and vertical jump performance are positively associated with match HIRD and sprint performance, respectively [20,35]. However, there are no published data on whether these relationships exist in NLF. To support the context of the match running speed demands, player physical capacities were assessed during the preseason. Contracted outfield players (n = 15) completed testing on a single day in a controlled laboratory setting after a standardised warm-up. This testing was part of a preseason assessment independent of the current study’s data collection. This included an aerobic capacity assessment of ${\dot{V}O}_{2\max}$ [36] via a maximal treadmill test using Vyntus One (Vyaire Medical, Hoechberg, Germany) with the Sentry Suite software package (version 2.21; Vyaire Medical, Hoechberg, Germany) and lower limb power assessed by countermovement jump (CMJ) height [37], measured with an optical beam measurement system (Optojump, Microgate, Bolzano, Italy).

### 2.3. Match Running Speed Demands

Match running speed data were collected using wearable GPS devices. Players used the same Polar Team Pro strap (Polar Electro, Kempele, Finland), and data were collected via the Polar Team app (Version 2.0, Polar Electro, Kempele, Finland), which is accurate and reliable for team sport movement tracking [38]. Participants’ data from 41 matches, comprising 311 observations (CD, n = 106; WD, n = 69; MF, n = 96; ST, n = 40), were included in the analysis. Only players completing the full 90 min of the match were included in the analysis. The position allocated to a player by the coach during a match may have differed from their primary playing position (e.g., a CD playing as a MF). The running speed demands were defined by thresholds per the literature [39]: walking (<7.1 km·h^−1^), jogging (7.2–14.3 km·h^−1^), running (14.4–19.7 km·h^−1^), high-speed running (19.8–25.1 km·h^−1^), and sprinting (>25.1 km·h^−1^).

### 2.4. Statistical Analysis

Descriptive analyses were completed in Microsoft Excel (Version 16.89.1, Microsoft Corporation, USA, 2021) and are displayed as (mean ± SD). The following analyses were computed in R [40]. A chi-square test of independence was used to assess the difference in match outcome (win, loss, draw) between coaches. Simple linear regression and linear mixed-effect (LME) models were computed using the “lme4” package [41].

A proposed model of running speed demands is outlined in the directed acyclic graph (DAG) (see Figure 1), which includes observed and unobserved variables. The model is based on the current literature [24,28,42,43,44], and relationships are associative rather than causal acknowledging the complexity and interdependency of factors. The model also explored potential explanatory factors that may influence running speed demands. The relationships included higher-level factors of coach, player/position factors (e.g., position, aerobic capacities), match outcome, situational variables (e.g., environmental, match location, fixture congestion) and lower-level technical/tactical factors. Variables for each match were coded for the respective Head Coach (Coach A, B, C), match outcome (win, lose, draw), player position (CD, WD, MF, ST), and running speed distance (walking, jogging, running, high-speed running, and sprinting distances).

Each running speed model included each of the Head Coach, player position, and match outcome variables, as well as match number as the random effect in mixed effects models. Including these variables allowed for a nuanced examination of running speed demands while capturing the impact of Head Coach change across the season. Modelling iterations began with simple univariate linear models of each variable and progressed to an LME model of coach, player position, match outcome, and random effect as match number. A visual inspection of the residual versus fitted plots and Q-Q plots was used to assess the linearity, homoscedasticity, and normality of residuals within models. To account for violations of assumptions within the models, robust standard errors (SEs) applying the “clubSandwich” package [45] were used in the interpretation of results [46], providing a more conservative estimate of variable significance.

The models were compared using the following model fit indices: Akaike information criterion (AIC), Bayesian information criterion (BIC), and root mean square error (RMSE). The multivariate mixed-effect model was superior to simple linear and multivariate models across each model fit metric and was used as the final model in the results. The model fit output and code are available in Appendix A.

## 3. Results

The preseason team average of outfield players’ (n = 15) ${\dot{V}O}_{2\max}$ was 57.4 ± 3.7 mL/kg/min and CMJ height was 42.2 ± 5.6 cm. The average number of matches played was 19.5 ± 12.5. The running speed metrics, detailed for the team overall and by position, are displayed in Table 1.

Across 41 matches, the distribution of match outcomes by Head Coach was as follows: Coach A = 26 matches (Won = 10, Drew = 7, Lost = 9), Coach B = 4 matches (Won = 1, Drew = 0, Lost = 3), and Coach C = 11 (Won = 6; Drew = 1, Lost = 4). The results showed no significant difference (χ2 = 4.373 (df = 4), *p* = 0.358).

The results of the running speed LME models using robust SE (± 95% confidence intervals) and model fit indices of AIC, BIC, and RMSE are displayed in Table 2. Each running speed LME model demonstrated a significant overall effect. There were significant negative associations in both TD and running models for Coach B (compared to reference Coach A). In contrast, Coach C exhibited a consistent negative association across all running speed models except for walking and sprinting. The most significant positional effects were observed in WD and MF positions (compared to reference CD), demonstrating positive associations in all running speed LME models except walking, which had a negative association, suggesting greater running speed demands of WD and MF positions. Match outcome had a weaker effect across running speed LME models, with only the HSR model demonstrating a positive association for both loss and draw outcomes (compared to reference win), and running model exhibiting a positive association for draw outcomes, signifying a weaker association between running speed demands and match outcome.

## 4. Discussion

This study provides novel insights into the running speed demands of professional NLF matches, focusing on positional differences and the impact of Head Coach changes. The results showed WD and MF players covered greater distances than central defenders and strikers, consistently with the existing literature in higher leagues [1,27,47]. Interestingly, Head Coach changes were associated with reductions in running speed demands, indicating running speed demands may be affected following these transitional periods.

The running performance of one professional NLF team over a single season revealed a team average TD of 10,479 ± 42 m per match, similarly to findings in professional (range 9–14 km) [1,11,18] and semi-professional NLF (10,163 ± 1183 m and 10,805 ± 158 m) [5,6]. Regarding positional differences in TD, WD and MF players covered the greatest distances, while ST and CD covered less. These positional differences in running speed demands align with the previous literature [13,27,48]. Contrastingly, significant disparities were noted in HSR and sprinting when compared to professional football and other NLF cohorts. The professional NLF cohort covered the following distances for HSR 431 ± 62 m and sprinting 99 ± 26 m. This is considerably lower than professional football players (618 to 1001 m for HSR and 153 to 295 m for sprinting) [10] and semi-professional NLF players (within the same tier), who covered 879 ± 50 m in HSR [6]. Therefore, these data suggest the current professional NLF cohort covered comparable HSR distances with players from amateur football (503 ± 198 m) [14]. It must be considered these data may also be influenced by different playing styles between semi-professional and professional NLF teams.

In English professional football, higher standards of play (i.e., the Premier League compared to the Championship and League 1) are associated with lower HIRD, suggesting that technical proficiencies may influence match play and running speed demands [27]. Similarly, this study found that the professional NLF cohort exhibited lower HIRD than a semi-professional NLF team [6]. The running speed demands may differ due to tactical or technical proficiencies rather than physical capacities alone. For instance, teams with more weekly training sessions and greater tactical preparation may require less high-intensity running during matches, resulting in lower HIRD, typically observed at higher football standards [49]. The lower running speed demands observed may reflect playing styles focused on possession and a greater emphasis on shorter passes, as professional NLF clubs try to emulate playing styles of elite teams. However, no data on technical proficiency within NLF have been published. While technical performances are associated with playing level and the success of a football club [11], other factors such as financial restrictions, limited equipment or lack of staff may impede team success in NLF [4]. This study provides preliminary evidence that the professional status of a club may influence running speed demands. However, it remains unknown whether the professional status of an NLF team translates to success within NLF, and thus, further research is warranted.

This study revealed that coach changes were associated with significant negative reductions in TD and running distances. Specifically, Coach B, an interim Head Coach, was associated with a lower TD and running LME models, while Coach C exhibited a consistent negative association across all running speed LME models except for walking and sprinting. This contrasts with observations from professional football, where, under a new coach, a team typically demonstrates increased physical performance [23]. For example, professional football teams usually show greater HIRD and/or TD following a coach change [33,50]. Similar effects have been observed in semi-professional football, with a greater TD and distance covered at high-speed accelerations but reduced HIRD [51]. However, uncertainty remains regarding the long-term relationship between a Head Coach change and resulting team success and running speed demands [23,32,33,50]. Despite lower overall running speed demands with a new Head Coach, there was no change in the distribution of match outcomes between coaches. This may be due to similar players available and the subsequent tactics employed by Head Coaches, as professional NLF clubs may be expected to play to their strengths and the advantages that full-time status lends to match preparation [52]. However, considering the differing skill levels of players in NLF and those in higher tiers of professional football, effective playing styles may be more specific to their level of play. Moreover, football clubs with long-standing coaches have been observed to accrue more points during a season [50], suggesting that stability may be beneficial.

The absence of transfer window restrictions in NLF increases the potential turnover of players and subsequent positional changes throughout the season. This may compound the effect of a new coach, further influencing the technical and tactical dynamics within a team. Players deployed outside their usual position may modify their style of play. Individual and tactical factors drive these consequent changes, and ultimately influence running speed profiles and technical proficiency [53]. Moreover, the variability in player experience in professional NLF teams, ranging from Premier League academy player to an experienced NLF semi-professional player, may influence how a team can adapt in periods of inconsistency and change. This study provides novel evidence that the impact of Head Coach change may differ in professional NLF from the typical increases in running speed demands observed in other football cohorts. Instead, factors such as understanding technical and tactical efficiencies may be more critical in NLF [4]. Consequently, a change in Head Coach when foundational aspects of footballing performance may not be fully addressed could constitute an unwise dismissal of the coach [54].

Match outcomes were associated with running speed demands in this study, with LME models revealing in matches lost and drawn, greater running and HSR distances were covered compared to those won. This finding is consistent with the existing literature on professional football, where players covered less distance in matches they won compared to those lost or drawn [47]. The state of outcome within a match, indicated by the score-line, may introduce a situational effect, influencing the tactics employed by a Head Coach and a motivational impact on a player’s running performance [28]. For instance, a draw may drive a greater running intensity to create goal scoring opportunities or press the opponent to regain possession to clinch success. The lower running and HSR distances observed in matches won, may suggest these matches demonstrate superior tactical execution (e.g., shots on target, effective defensive pressing) and associated running performance, which are typically associated with match success [44]. Teams at higher league levels consistently demonstrate more defensive running during matches, contributing to match success [55]. However, the tactical behaviours observed within NLF may have inconsistencies that reflect differing levels of play, and the role of GPS data and performance analysis may provide a critical differential in understanding tactical nuances. While running speed demands contribute to match performance, technical and tactical behaviours influence match outcomes more [44]. Often these tactical actions, such as pressing or counter attacking to create goal scoring opportunities, are driven by high-intensity actions [42,49]. Moreover, high-intensity running decreased as the season progressed despite the three different Head Coaches, which may indicate a lack of application of GPS data to inform match tactics.

In this study, the ${\dot{V}O}_{2\max}$ of professional NLF players (57.4 ± 3.7 mL/kg/min) was generally comparable to values observed across amateur to elite football cohorts but lower than that typically seen in professional league players (58.2 to 62.2 mL/kg/min), and more consistent with that of amateur players (57.8 to 61.7 mL/kg/min) [56]. This may have impacted the TD and HIRD observed in this study, as lower-level players typically demonstrate both an inferior aerobic capacity and a lower TD covered during matches [11,17,20]. However, the CMJ height (42.2 ± 5.6 cm) was similar to other observed professional cohorts [35,43], suggesting that lower limb power is sufficient to meet the physical demands of match play related to sprint performance [17,35]. The GPS data may support NLF clubs to monitor and evaluate players’ running performance [15], and allow for correlation with physical testing (i.e., CMJ and ${\dot{V}O}_{2\max}$). Further investigation of NLF players’ physical capacities and match running speed demands is needed, as various professional leagues have attributed greater variation in match running speed demands to tactics and playing styles [11,26]. Clubs operating in NLF should critically evaluate match GPS data, if staffing and resource constraints allow, and consider the implications for training needs of players and the possible impact on match success.

Fitness testing and training monitoring practices in NLF appear inconsistent, with barriers such as limited staff and time constraints hindering implementation [4]. This lack of systematic testing or monitoring may contribute to the lower aerobic capacity of professional NLF players and make evaluating and understanding team performance difficult. While ${\dot{V}O}_{2\max}$ testing or other aerobic tests, such as the Yo-Yo intermittent endurance test 2 (IE2), may not be specific to the demands during a football match [36], associations have been observed between Yo-Yo IE2 performance and HIRD in professional football [27]. Furthermore, a higher ${\dot{V}O}_{2\max}$ has been shown to distinguish higher-level players from their lower-level counterparts [56]. Professional NLF teams may have more time to dedicate to fitness than semi-professional teams, which has been demonstrated to positively impact match performance when implemented correctly [57].

### Limitations and Future Directions

This study provides novel longitudinal data on the running speed demands of professional NLF players and the influence of a Head Coach change at this level. However, several limitations must be acknowledged. The primary limitation is the sample size, as data were collected from one NLF professional football club across a single season, limiting the external validity and transferability to other professional NLF cohorts. Additionally, the anonymised player data restricted the analysis to only outfield players completing a full match to be included and may affect generalisability to players with partial match involvement (i.e., <90 min). The data are from the 2018–2019 season and may not be representative of the current running speed demands but provide NLF clubs with key insights into training strategies. Due to the dynamic nature of football and coaching strategies, players may have played in a different position than their allocated position in some matches. Future studies with larger samples could provide further analysis on the positional and formation effects on running speed demands. Teams competing in NLF are not limited to transfer windows, and it is usual for players to join during the season, so certain coaches may recruit more players, and thus, future studies may consider to account for squad variability across a season. Furthermore, only full matches were analysed, and greater insights could be gained by examining the running speed demands of each match half separately.

This study’s conceptual model, illustrated in the DAG, depicts contextual-level variables hypothesised to influence running speed demands. The simplistic model does not imply direct causality, highlighting potential factors that warrant further exploration using more sophisticated frameworks. The LME model used is suited to handle unbalanced observation counts, although the small sample sizes of variables (e.g., Coach B) may have affected the robustness of the model estimates. The prioritisation in data analysis was on the interpretability of results rather than strategies to mitigate these limitations. Nonetheless, to enhance robustness, we employed robust standard errors to address potential violations of model assumptions, assessed model fit indices, and grounded the model in established theoretical frameworks.

The unpredictable nature of NLF, including frequent player and coach changes throughout a season, may have contributed to inconsistencies across team-level factors. Future research should aim to capture larger samples across multiple clubs and additional seasons to improve external validity. A greater understanding of match demands may lead to performance improvements and inform match or training strategies relative to this level. For instance, insight into technical proficiency during matches and the mechanisms of play leading to goal opportunities could prove invaluable to NLF clubs and practitioners.

## 5. Conclusions

This is the first study to investigate the competitive match running speed demands and the impact of a Head Coach change in professional NLF. The findings demonstrate comparable average TD and positional differences with other professional football and NLF cohorts. However, significant discrepancies are noted in the high-intensity running speeds compared to the ranges typically observed in professional football and other semi-professional NLF cohorts. These differences may reflect variations in aerobic fitness or teams’ technical and tactical preparedness at this level.

Interestingly, in contrast to the other literature, a Head Coach change in a professional NLF team was associated with a reduction in running speed demands, with no significant improvement in match outcomes. This may reflect influential factors beyond physical performance, such as the challenges in implementing tactical approaches suited to a professional team that may exceed technical or physical capabilities at this level. This may also highlight the broader difficulties in meeting the expectations of a professional team where, unlike their higher league counterparts, NLF teams are faced with limited resources and club staffing, and player experience can vary significantly.

The Polar Team Pro GPS was a sufficiently reliable and accurate solution for capturing match external loads in professional NLF setting. In this study, the GPS data enabled detailed motion-analysis, to profile position-specific running performances and capture the effects of Head Coach changes. These findings enable benchmarks for conditioning and training prescription. However, the main limitation to the practical application of GPS within NLF remains operational, with limited staffing that can affect the integration of running speed data into evidenced-based coaching decisions.

The disparities observed in this study highlight the unique challenges NLF clubs face for a successful transition to league football. Teams operating at this level with access to GPS devices should consider the team and position specific running speed demands, and clubs should provide suitable operational support to effectively apply these findings to training practices and coaching strategies. These findings emphasise the need to further explore running speed demands, technical and tactical factors, and other determinants that may influence success in NLF.

## Figures and Tables

**Figure 1 sensors-25-02865-f001:**
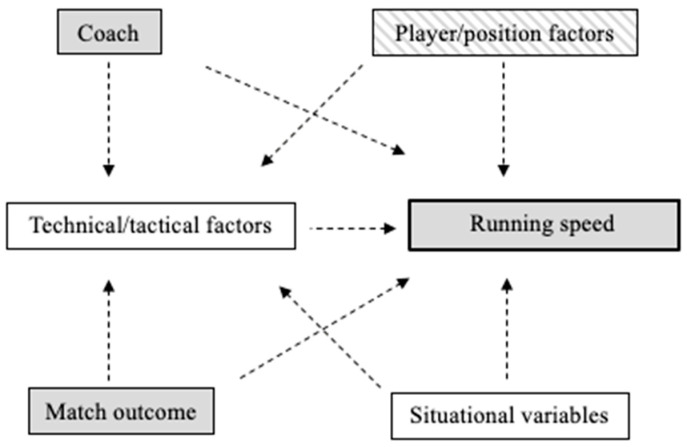
Directed acyclic graph of observed (grey), unobserved (white), and partially observed (striped) variables in conceptual running speed demands model.

**Table 1 sensors-25-02865-t001:** Season running speed demands by position and team average. Data are displayed as mean ± SD.

	CD	WD	MF	ST	TEAM
TD (m)	10,211 ± 373	10,602 ± 476	10,714 ± 469	10,457 ± 447	10,479 ± 421
Walking (m) (<7.1 km·h^−1^)	4084 ± 139	3836 ± 129	3937 ± 163	4020 ± 119	3972 ± 140
Jogging (m) (7.2–14.3 km·h^−1^)	4293 ± 237	4290 ± 273	4597 ± 286	4356 ± 251	4398 ± 260
Running (m) (14.4–19.7 km·h^−1^)	1567 ± 168	1594 ± 189	1612 ± 212	1508 ± 158	1578 ± 194
HSR (m) (19.8–25.1 km·h^−1^)	334 ± 41	511 ± 75	472 ± 68	457 ± 57	431 ± 62
Sprinting (m) (>25.1 km·h^−1^)	50 ± 17	155 ± 41	116 ± 116	97 ± 24	99 ± 26

Note. HSR = high-speed running; CD = central defender; WD = wide defender; MF = midfielder; ST = striker.

**Table 2 sensors-25-02865-t002:** Summary of linear mixed-effect models for running speed demands: effects of Head Coach change, position, and match outcome.

	**Intercept**	Coach B	Coach C	Position MF	Position ST	Position WD	Outcome D	Outcome L	*F*	*p*	AIC	BIC	RMSE
**Total** **Distance (m)**	**10,393.93** (10,190.89, 10,596.97)	**−364.61** (−724.82, −4.39)	**−505.31** (−716.82, −293.80)	**410.32** (248.78, 571.87)	−155.35 (−393.95, 83.24)	**560.44** (402.20, 718.67)	86.54 (−242.37, 415.45)	25.15 (−223.47, 273.77)	11.41	<0.001	4839	4876	586
**Walking (m)** **(<7.1 km·h^−1^)**	**3986.14** (3901.71, 4070.57)	59.07 (−67.08, 185.23)	67.77 (−35.75, 171.28)	**−77.84** (−151.25, −4.42)	**141.24** (12.15, 270.34)	**−184.32** (−248.94, −119.70)	54.39 (−75.70, 184.48)	−8.88 (−104.16, 86.40)	4.98	0.002	4374.	4412	282
**Jogging (m)** **(7.2–14.3 km·h^−1^)**	**4466.97** (4333.78, 4600.17)	−194.10 (−395.83, 7.62)	**−200.01** (−364.83, −35.19)	**153.79** (30.27, 277.30)	**−442.67** (−629.51, −255.82)	**357.77** (211.34, 504.21)	−134.15 (−308.83, 40.52)	−97.47 (−262.23, 67.29)	8.70	<0.001	4713	4751	500
**Running (m)** **(14.4–19.7 km·h^−1^)**	**1551.63** (1456.69, 1646.56)	**−169.16** (−319.73, −18.60)	**−297.79** (−373.12, −222.47)	**160.95** (73.96, 247.94)	**−106.23** (−191.94, −20.52)	**145.95** (65.94, 225.96)	**111.77** (5.97, 217.56)	84.24 (−6.18, 174.67)	19.15	<0.001	4372	4410	287
**HSR (m)** **(19.8–25.1 km·h^−1^)**	**327.37** (293.70, 361.03)	−42.28 (−105.28, 20.72)	**−57.93** (−94.59, −21.28)	**122.89** (94.69, 151.08)	**143.52** (101.80, 185.24)	**168.96** (134.30, 203.62)	**60.21** (13.43, 106.98)	**42.87** (5.88, 79.86)	19.74	<0.001	3800	3837	111
**Sprinting (m)** **(>25.1 km·h^−1^)**	**50.66** (38.72, 62.59)	−8.27 (−38.66, 22.12)	−10.12 (−25.20, 4.95)	**42.24** (29.67, 54.81)	**106.90** (84.61, 129.18)	**75.69** (57.16, 94.21)	7.87 (−7.84, 23.57)	15.95 (−2.16, 34.06)	21.13	<0.001	3364	3402	55

Note: HSR = high-speed running; MF = midfielder; ST = striker; WD = wide defender; AIC = Akaike information criterion; BIC = Bayesian information criterion; RMSE = root mean square error. Significant variables (95% CI not crossing zero) are shown in bold. Reference groups = Coach A, CD, Outcome W.

## Data Availability

For information regarding the R code and statistical analysis, please see Appendix A. All relevant team data are included within the study, and further requests are welcomed by the authorship team.

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
