# Peer review of "Competitive Match Running Speed Demands and Impact of Changing the Head Coach in Non-League Professional Football†"

_sensors, 2025, doi:10.3390/s25092865_

Round 1

Reviewer 1 Report

Comments and Suggestions for Authors

The current study being reviewed is a longitudinal cohort study examining the competitive match running speed demands and impact of changing the head coach in non-league professional football. The authors completed a well-executed prospective longitudinal cohort study. The authors found significant positional differences with high-speed running and sprinting distances significantly lower in professional NLF compared to other football cohorts. Additionally, changing the coach mid-season had a significant impact on these numbers, reducing total distance and other running metrics.

This is an interesting, well-done study and would contribute to the literature. This reviewer does not have any significant edits prior to acceptance for submission to Sensors.

Author Response

The current study being reviewed is a longitudinal cohort study examining the competitive match running speed demands and impact of changing the head coach in non-league professional football. The authors completed a well-executed prospective longitudinal cohort study. The authors found significant positional differences with high-speed running and sprinting distances significantly lower in professional NLF compared to other football cohorts. Additionally, changing the coach mid-season had a significant impact on these numbers, reducing total distance and other running metrics.

This is an interesting, well-done study and would contribute to the literature. This reviewer does not have any significant edits prior to acceptance for submission to Sensors.

Authors response: Many thanks for your comments and feedback. We appreciate your time taken to review our manuscript.

Reviewer 2 Report

Comments and Suggestions for Authors

Review / Competitive Match Running Speed Demands and Impact of Changing the Head Coach in Non-League Professional Football

Introduction:

The second and third paragraphs should be reorganized – First introduce GPS devices, then you can speak about demands (extensity and intensity) and then about differences between positions, playing level and similar.

As NLF clubs can be professional or semi-professional, there may be differences in training time and player demands (e.g.,employment other than football), potentially influencing players’ aerobic and anaerobic capacities and, consequently, running speed demands compared to semi-professional  teams. For example, in professional football, V̇O2max was positively associated with HIRD during matches [10].“ – Player demands syntagm can mislead to the demands on the football pitch (although you put an explanation in the bracket. Please use another term. Also, the last sentence stands alone and is missing additional explanation and/or context.

For instance, in English professional football, PL clubs, which contain the most technically proficient teams... „ – You probably meant the „most technically proficient player“.

A change in Head Coach can also affect the running speed demands during matches“ – Explain why

Methods

Why did you choose only 4 playing positions? What about wide forwards? Their demands differ from those of both midfielders and strikers.

How many game performances you analysed per each positon?

How are vo2 max and power testing important in the context of your investigation?

It is unclear what exact variables you analyzed. Please clarify.

Discussion

The running speed demands of professional NLF revealed TD of 10,479 ± 42 m per match, similar to findings in professional (range 9-14 km).“ – You need to highlight that this is the average value per player. Also replace „running speed demands“ with „running performance“.

This indicates the observed TD is representative of other NLF studies.“ – which other NLF studies?

When comparing your results with the other studies you need to take into consideration that you observed only one team and that this could effect the results.

Provide a more detailed explanation for your results regarding the association between running performance and match outcome.

I don’t understand the importance and relation of physical components (vo2 max and CMJ) and running performance. This paragraph looks like it belongs to some other article.

Conclusion

What is the importance of your findings?

What are the practical implications?

Finally, why did you choose Sensors journal as your manuscript refers to specific game demands and there is no word about specific sensors or measuring equipment?

Author Response

Introduction:

The second and third paragraphs should be reorganized – First introduce GPS devices, then you can speak about demands (extensity and intensity) and then about differences between positions, playing level and similar.

As NLF clubs can be professional or semi-professional, there may be differences in training time and player demands (e.g.,employment other than football), potentially influencing players’ aerobic and anaerobic capacities and, consequently, running speed demands compared to semi-professional  teams. For example, in professional football, V̇O2max was positively associated with HIRD during matches [10].“ – Player demands syntagm can mislead to the demands on the football pitch (although you put an explanation in the bracket. Please use another term. Also, the last sentence stands alone and is missing additional explanation and/or context.

For instance, in English professional football, PL clubs, which contain the most technically proficient teams... „ – You probably meant the „most technically proficient player“.

A change in Head Coach can also affect the running speed demands during matches“ – Explain why

Methods

Why did you choose only 4 playing positions? What about wide forwards? Their demands differ from those of both midfielders and strikers.

How many game performances you analysed per each positon?

How are vo2 max and power testing important in the context of your investigation?

It is unclear what exact variables you analyzed. Please clarify.

Discussion

The running speed demands of professional NLF revealed TD of 10,479 ± 42 m per match, similar to findings in professional (range 9-14 km).“ – You need to highlight that this is the average value per player. Also replace „running speed demands“ with „running performance“.

This indicates the observed TD is representative of other NLF studies.“ – which other NLF studies?

When comparing your results with the other studies you need to take into consideration that you observed only one team and that this could effect the results.

Provide a more detailed explanation for your results regarding the association between running performance and match outcome.

I don’t understand the importance and relation of physical components (vo2 max and CMJ) and running performance. This paragraph looks like it belongs to some other article.

Conclusion

What is the importance of your findings?

What are the practical implications?

Finally, why did you choose Sensors journal as your manuscript refers to specific game demands and there is no word about specific sensors or measuring equipment?

Authors response: Many thanks for the time spent reviewing our manuscript, and for your thoughtful and constructive feedback. We believe our manuscript is improved following your comments and the changes from your recommendations. Please see individual responses to your points and the corresponding highlighted changes within text.

________________________________________________________

Introduction:

The second and third paragraphs should be reorganized – First introduce GPS devices, then you can speak about demands (extensity and intensity) and then about differences between positions, playing level and similar.

Authors response: Many thanks for your comment, please see amended paragraph structure.

Additional detail has also been added to the GPS paragraph: “The Polar Team Pro (Polar Electro, Kempele, Finland) system provides relatively inexpensive access to a GPS and accelerometer unit, with a sampling rate of 10 Hz, for capturing match physical performance data [57]. These data may be used for performance analysis or to inform on injury risk [58]. However, their utility and application of GPS data and performance analysis within NLF remains unclear, and may vary from professional contexts due to staffing and financial constraints [4].” (LINES 60-65)

As NLF clubs can be professional or semi-professional, there may be differences in training time and player demands (e.g.,employment other than football), potentially influencing players’ aerobic and anaerobic capacities and, consequently, running speed demands compared to semi-professional  teams. For example, in professional football, V̇O2max was positively associated with HIRD during matches [10].“ – Player demands syntagm can mislead to the demands on the football pitch (although you put an explanation in the bracket. Please use another term. Also, the last sentence stands alone and is missing additional explanation and/or context.

Authors response: Thank for your comments, the term “player demands” has been replaced with “off-field commitments” to avoid any ambiguity conferred. The final sentence has been moved to provide context for inclusion of VO2max and CMJ variables within this study.

For instance, in English professional football, PL clubs, which contain the most technically proficient teams... „ – You probably meant the „most technically proficient player“.

Authors response: Many thanks for your thorough proof reading, this is correct. The wording has been updated accordingly.

A change in Head Coach can also affect the running speed demands during matches“ – Explain why

Authors response: Many thanks for your comments, please see amended content: “A change in Head Coach can also affect the running speed demands during matches, with short term increases in running performance attributed to the motivational improvement of the team [30,31].” (LINES 108-109)

Methods

Why did you choose only 4 playing positions? What about wide forwards? Their demands differ from those of both midfielders and strikers.

Authors response: Thank you for your comment, we have added context of the formation, with these four positions used in a 3-5-2 formation across the season: “Player positions were assigned by the Head Coach, utilising a 3-5-2 tactical formation predominantly throughout the season. These positions were defined as central strikers (ST)…” (LINES 147-148)

An additional comment was also stated in the limitations: “Future studies with larger samples could provide further analysis on the positional and formation effects on running speed demands.” (LINES:383-384)

How many game performances you analysed per each positon?

Authors response: Thank you for your comment, there were a total of 106, 69, 96 and 40 for CD, WD, MF and ST, respectively. The manuscript has been updated (LINE:171)

How are vo2 max and power testing important in the context of your investigation?

Authors response: Many thanks for your comment, we have expanded on the reason to include these physical components. Within professional and semi-professional football, relationships between these metrics and running volume has been observed. It was hypothesised that similarly, a relatively high VO2 max and jump performance would correlate with running performance in keeping with the literature. This association and importance of GPS and physical testing has been further discussed within the discussion.

Context has been expanded within the introduction (LINES 70-73).

The methods now includes: “In professional football, both V̇O2max and vertical jump performance are positively associated with match HIRD and sprint performance, respectively [10,54]. However, there is no published data on whether these relationships exist in NLF.” (LINES 154-156)

It is unclear what exact variables you analyzed. Please clarify.

Authors response: Thank you for your comment and insight on the interpretability of our methods. The full list of variables is now outlined (LINES 191-194). The running speed models are then detailed with reference to all variables analysed (LINES 200-202).

Discussion

The running speed demands of professional NLF revealed TD of 10,479 ± 42 m per match, similar to findings in professional (range 9-14 km).“ – You need to highlight that this is the average value per player. Also replace „running speed demands“ with „running performance“.

Authors response: Thank you for your comment, the manuscript has been updated accordingly.

This indicates the observed TD is representative of other NLF studies.“ – which other NLF studies?

Authors response: Thank you for your comment, this sentence refers to the prior sentence …” semi-professional NLF football (10,163 ± 1183 m and 10,805 ± 158 m) [5-6]. This indicates the observed TD is representative of other NLF studies”.

The sentence has been removed as it added no further context and may be an oversight to state the TD is representative of NLF citing 2 studies.

When comparing your results with the other studies you need to take into consideration that you observed only one team and that this could effect the results.

Authors response: Many thanks for your comment, this has now been highlighted at the beginning of the discussion (LINES 256-257) and also stated within limitations (LINE 375-376).

Provide a more detailed explanation for your results regarding the association between running performance and match outcome.

Authors response: We appreciate your constructive feedback and have now expanded our discussion on your recommendation. Please see the updated paragraph (LINES 321-342).

I don’t understand the importance and relation of physical components (vo2 max and CMJ) and running performance. This paragraph looks like it belongs to some other article.

Authors response: Thank you for your comment, the physical testing and monitoring of players within NLF remains unclear. There are known associations in football matches between physical capacities (vo2max and CMJ) and running performance. However, there is no data for non-league level. Importantly, this GPS data highlights a performance gap in running performance which correlates with lower physical capacity. Due to lower staffing and resources at this level, this performance gap may be more difficult to detect with varying sports science support and practices. Please see updated paragraph in text to reflect these changes (LINES 343-358)

Conclusion

What is the importance of your findings?

Authors response: thank you for your comment. This is the first study to investigate match running speed demands of NLF and the effects of Head Coach change on the running speed demands, as well as providing insights into positional differences at this level. Highlights areas for improvement on training implications. The conclusion is more clearly structured to highlight the importance of the study findings, we believe your suggestion has improved the quality and interpretation of the study findings.

What are the practical implications?

Authors response: Many thanks for your comment. Due to the heterogenous nature of NLF and variation between clubs, generalised guidance has been suggested to teams in the closing paragraph (LINES 432-435). We appreciate your thoughtful feedback and recommendations throughout our manuscript, and believe it is now improved following our changes.

Finally, why did you choose Sensors journal as your manuscript refers to specific game demands and there is no word about specific sensors or measuring equipment?

Authors response: Thank you for your comment. This manuscript provides a unique perspective in football and sub-elite levels of sport; and contains novel data for practitioners at this level. It captures both the challenges and application of GPS systems for analysing match performance and how this may inform training demands. We have also now included a summary paragraph on the application of GPS to this level within the conclusion (LINES 424-430).

The authorship team believes our manuscript is suitable for this journal, with specific applicability to this special issue, which aims to gather research on technologies, applications, and challenges in the field of inertial sensing system for motion monitoring. The manuscript now contains updated sensor and device information, to provide context and comparability for other research at this level.

Reviewer 3 Report

Comments and Suggestions for Authors

Very interesting results appeared to be difficult to interpret. It seems important to conduct a study where the coach would not change during the season and possibly compare the two studies. In addition, from the results obtained, it seems that despite the change in running parameters, similar match results were obtained, which seems difficult to interpret. As for the different effects of training and the obtained running parameters after changing the coach, according to the reviewer, the expertise and personality of the coach, i.e. his individual qualities, are of the greatest importance.

The conclusions should include an assessment of the usefulness of GPS devices used to make the study. 

Author Response

Very interesting results appeared to be difficult to interpret. It seems important to conduct a study where the coach would not change during the season and possibly compare the two studies. In addition, from the results obtained, it seems that despite the change in running parameters, similar match results were obtained, which seems difficult to interpret. As for the different effects of training and the obtained running parameters after changing the coach, according to the reviewer, the expertise and personality of the coach, i.e. his individual qualities, are of the greatest importance.

The conclusions should include an assessment of the usefulness of GPS devices used to make the study.

Authors response: Many thanks for your comments and feedback. We appreciate the time you have taken to review our manuscript. Based on your recommendation, we have updated the manuscript and included an evaluation of the application of GPS devices in the study (LINES 424-430), and believe the manuscript is improved with your suggestion.

Round 2

Reviewer 2 Report

Comments and Suggestions for Authors

Thank you for reviewing the manuscript according my instructions. Congrats on your work!